# THE CODING LIMITS OF ROBUST WATERMARKING FOR GENERATIVE MODELS

## ABSTRACT

We prove a sharp threshold for the robustness of cryptographic watermarking for generative models. This is achieved by introducing a coding abstraction, which we call *messageless secret-key codes*, that formalizes sufficient and necessary requirements of robust watermarking: soundness, tamper detection, and pseudorandomness. Thus, we establish that robustness has a precise limit: For binary outputs no scheme can survive if more than half of the encoded bits are modified, and for an alphabet of size $q$ the corresponding threshold is $(1 - 1/q)$ of the symbols.

Complementing this impossibility, we give explicit constructions that meet the bound up to a constant slack. For every $\delta > 0$, assuming pseudorandom functions and access to a public counter, we build linear-time codes that tolerate up to $(1/2)(1-\delta)$ errors in the binary case and $(1-1/q)(1-\delta)$ errors in the $q$-ary case. Together with the lower bound, these yield the maximum robustness achievable under standard cryptographic assumptions.

We then test experimentally whether this limit appears in practice by looking at the recent watermarking for images of Gunn, Zhao, and Song (ICLR 2025). We show that a simple crop and resize operation reliably flipped about half of the latent signs and consistently prevented belief-propagation decoding from recovering the codeword, erasing the watermark while leaving the image visually intact.

These results provide a complete characterization of robust watermarking, identifying the threshold at which robustness fails, constructions that achieve it, and an experimental confirmation that the threshold is already reached in practice.

## 1 INTRODUCTION

In recent years, generative AI models such as GPT, Llama, and Claude have made it increasingly difficult to tell human-produced content from machine-generated text, code, and images. This new reality raises a central question for both technology and society: how can we reliably distinguish what is AI-generated from what is authentically human?

Watermarking is one of the most promising strategies for meeting this challenge (Executive Office of the President of the United States, 2023; European Parliament and Council of the European Union, 2024). The idea is to embed a secret, imperceptible pattern into model outputs at generation time, enabling later verification by anyone with the appropriate key. Ideally, a watermark should be robust, surviving even substantial adversarial editing, while also being undetectable to those without the key and leaving the original content unchanged in quality or meaning (Aaronson, 2022; Kirchenbauer et al., 2023; Christ & Gunn, 2024).

Recent work by Christ & Gunn (2024) formalized cryptographic watermarking using pseudorandom error-correcting codes (PRCs), showing that such watermarks can be both undetectable and robust to significant amounts of tampering. But their results naturally lead to a deeper question: what are the true limits of robustness for any cryptographically grounded watermark? Is there a universal barrier that no watermarking scheme, however cleverly designed, can overcome?

This question has been explored from two sides. On one side, the "Watermarks in the Sand" (WiTS) work (Zhang et al., 2023; 2024) showed that any sufficiently robust watermark is, in principle, vulnerable: if an adversary has enough power, especially access to quality and perturbation oracles, it can eventually erase any watermark without degrading content quality. However, WiTS leaves open

the concrete thresholds for robustness that efficient watermarking schemes can actually achieve. On the other side, PRC-based schemes demonstrate the power of current cryptographic techniques but do not establish whether their level of robustness is the best possible.

In this paper, we bring these perspectives together. We introduce a conceptually simple abstraction, the *messageless secret-key code*, which crystallizes the core challenges and possibilities of cryptographic watermarking. Using this notion, we prove a tight information-theoretic threshold: no watermark, regardless of computational assumptions, can reliably survive if more than half of the encoded bits are altered in the binary case. More generally, for a $q$-ary alphabet, the limit is $(1-1/q)$ of the encoded symbols. Conversely, for any small constant $\delta$, we give explicit constructions that approach these limits, providing robustness up to just under half of the bits (for binary) or just under $(1 - 1/q)$ (for $q$-ary) of the symbols changed.

To make our results concrete, we analyze a recent state-of-the-art PRC-based watermarking scheme for images by Gunn et al. (2025). We show that a simple crop-and-resize operation, one that visually preserves the image but changes about half of its latent bits, is sufficient to reliably erase the watermark in practice. This demonstrates that the theoretical limit we establish is not just a worst-case artifact, but a real constraint in practical watermarking. This attack exposes a striking contrast between modalities: in text, watermark removal requires extensive edits that inevitably alter content, while for images, even a single benign transformation can erase the watermark without changing the image in any meaningful way. This makes the problem of robust watermarking in images uniquely delicate, as the watermark can disappear without a trace and without a visible cost.

Our findings position the impossibility results of WiTS in a precise, quantitative framework: WiTS demonstrates that all robust watermarks are eventually breakable in the black-box oracle model, while our work identifies the exact numerical threshold at which robustness becomes fundamentally impossible for any PRC-style, cryptographically undetectable watermark.

Looking forward, our results highlight that any significant increase in watermark robustness will require fundamentally new ideas, potentially leveraging semantic or structural features of content rather than cryptographic pseudorandomness alone.

## 1.1 OUR CONTRIBUTIONS

Our work provides a definitive and technically sharp characterization of the possibilities and limitations of cryptographic watermarking. Specifically, our contributions are as follows:

1. **A Simplified Abstraction for Watermarking.** We introduce and formalize *messageless secret-key codes* (also referred to as "zero-bit" in prior literature), a cryptographic primitive that captures the minimal requirements for watermarking generative models. These codes do not encode explicit messages but instead focus on pseudorandomness, soundness, and robust tamper-detection. We prove that messageless codes are not only sufficient but also necessary for cryptographic watermarking of generative AI outputs, even when the watermark detector has access to the prompt.

2. **Optimal Constructions.** We provide an explicit and efficient construction of messageless secret-key codes under the standard assumption of secure pseudorandom functions and the existence of an untamperable public counter. This construction achieves robustness up to nearly half the codeword symbols, as formalized below. Our construction also supports codes over larger alphabets.

   **Theorem 1** (Optimal Messageless Code, Informal). *For any constant $\delta > 0$ and any alphabet of size $q \geq 2$, there exists a simple, explicit construction of a messageless secret-key code (using a PRF and untamperable counter) that achieves tamper-detection robustness against adversarial modification of fewer than the threshold $\alpha^*$ fraction of codeword symbols where*

$$\alpha^* = \begin{cases} \frac{1}{2}(1-\delta), & \text{for binary alphabets;} \\ (1-1/q)(1-\delta), & \text{for } q\text{-ary alphabets.} \end{cases} \tag{1}$$

3. **Tight Information-Theoretic Limits.** We prove a matching lower bound: no messageless secret-key code (and hence no cryptographic watermarking scheme, even with negligible soundness and tamper-detection error) can robustly detect tampering if an adversary can modify more than the threshold $\alpha^*$ (as defined in (1) above) fraction of the symbols, for any constant $\delta > 0$. This impossibility holds even if the watermark detector is provided the prompt, and applies to both secret-key and public-key versions of the primitive.

4. **Optimality of PRC-Based Watermarking.** Our upper and lower bounds show that the robustness achieved by the PRC watermarking schemes of Christ and Gunn (Christ & Gunn, 2024) is in fact information-theoretically optimal. Any cryptographic watermark (binary or $q$-ary, secret-key or public-key) cannot exceed these limits.

5. **Concrete Attack.** We demonstrate that these theoretical limits are tight and immediately relevant in practice. Specifically, we show that a simple crop-and-resize transformation modifies enough underlying latent bits to consistently and silently remove the state-of-the-art PRC watermark (Gunn et al., 2025), without visible quality loss.

   Earlier work had already shown that watermarks can in principle be erased. WiTS established this with oracle access, showing that by repeatedly regenerating high-quality outputs an adversary can eventually reach an unmarked image (Zhang et al., 2023; 2024). More recently, the UnMarker attack (Kassis & Hengartner, 2024) demonstrated removal in a fully black-box setting, but only by perturbing essentially every pixel through computationally heavy spectral optimization, leaving behind small but measurable degradation.

   What distinguishes our result is that no such machinery is required. A single, routine crop-and-resize suffices to drive the system exactly to the threshold where our lower bound guarantees failure. In this sense, the limit is not just a theoretical boundary but a practical one, realized by the most ordinary of edits.

Collectively, these results establish the precise information-theoretic boundary for cryptographic watermark robustness, even under generous adversarial models. They show both what is achievable, and what is fundamentally impossible, with cryptographic watermarking alone.

## 1.2 RELATED WORK

To meet space constraints, the related work is deferred to Appendix A.1.

## 2 PRELIMINARIES ON WATERMARKING FOR GENERATIVE MODELS

For the notation used in this work, we refer the reader to Appendix A.2. Below, we recall the formalization of watermarking for generative models, following Zhang et al. (2023; 2024).

A generative model is any (possibly randomized) algorithm that, given a prompt $x$ (such as a question or a text description), produces an output $y$ (such as text or an image).

**Definition 1** (Generative models). A conditional generative model Model : $\mathcal{X} \to \mathcal{Y}$ is a probabilistic polynomial-time (PPT) algorithm that, given a prompt $x \in \mathcal{X}$, outputs $y \in \mathcal{Y}$. Here, $\mathcal{X}$ is the *prompt space*, and $\mathcal{Y}$ is the *output space*. We write $y \leftarrow_\$ \mathsf{Model}(x)$ to denote sampling a response from the model on prompt $x$.

A watermarking scheme is a systematic way to mark the outputs of a generative model, so that later a verifier with the appropriate key can detect whether a given output is watermarked. We focus on *secret-key* watermarking: both embedding and detection require a secret key.

A watermarking scheme $\Pi = (\mathsf{Watermark}, \mathsf{Detect})$ for a family of generative models $\mathcal{M} = \{\mathsf{Model}\}$ consists of two efficient algorithms:

$\mathsf{Watermark}(1^\lambda, \mathsf{Model})$: On input a security parameter $1^\lambda$ and a model Model, the algorithm outputs a secret key $\kappa \in \mathcal{K}$ and a watermarked model $\mathsf{Model}_\kappa : \mathcal{X} \to \mathcal{Y}$.

$\mathsf{Detect}(\kappa, x, y)$: On input a secret key $\kappa$, a prompt $x \in \mathcal{X}$, and an output $y \in \mathcal{Y}$, this algorithm returns either true or false, indicating whether $y$ is watermarked (in response to $x$).

A robust watermarking scheme should satisfy three key properties. First, **correctness**: watermarked outputs should always be recognized as such (the formal definition is given in Appendix A.3). Second, **soundness**: non-watermarked outputs (such as human-written or independently generated model outputs) should almost never be mistakenly detected as watermarked. Third, **robustness**: the watermark should survive small modifications to the output. That is, an adversary cannot erase the watermark by making limited edits unless a large fraction of the output is changed.

**Definition 2** (Soundness of watermarking). Let $\mathcal{M} = \{\mathsf{Model} : \mathcal{X} \to \mathcal{Y}\}$ be a class of generative models. We say that $\Pi$ satisfies *soundness* if for every $\mathsf{Model} \in \mathcal{M}$, every $x \in \mathcal{X}$, and every $y \in \mathcal{Y}$, $\mathbb{P}\big[\mathsf{Detect}(\kappa, x, y) = \mathtt{true} : (\kappa, \mathsf{Model}_\kappa) \leftarrow_\$ \mathsf{Watermark}(1^\lambda, \mathsf{Model})\big] \leq \mathsf{negl}(\lambda)$.

Robustness requires that a watermark remain detectable even if an adversary modifies a small part of the output. To capture this, we consider *tampering functions* $f : \Sigma^n \to \Sigma^n$ which, for a fixed parameter $\alpha \in (0, 1)$, output a string of the same length as the input and differ from the original output $y \in \Sigma^n$ in at most $\alpha n$ positions. Here, $\alpha$ specifies the maximum allowable fraction of edits.

We study two models of adversarial tampering: (1) In the *arbitrary tampering* model, the adversary may choose any positions and values for up to $\alpha n$ changes, allowing fully coordinated edits. (2) In the *independent tampering* model, the adversary changes each symbol independently, subject to the same overall budget $\alpha n$ (e.g., each symbol is flipped with probability at most $\alpha$).

Let $\hat{\mathcal{F}}_\alpha$ denote the set of (possibly randomized) functions $f$ as above (arbitrary), and let $\hat{\mathcal{F}}_\alpha^{\mathrm{ind}} \subseteq \hat{\mathcal{F}}_\alpha$ denote those which act independently on each symbol.

**Definition 3** ($\hat{\mathcal{F}}$-Robustness of watermarking). Let $\mathcal{M} = \{\mathsf{Model} : \Sigma^* \to \Sigma^*\}$ be a class of generative models. A watermarking scheme $\Pi$ is $\hat{\mathcal{F}}$-*robust* if for every $\mathsf{Model} \in \mathcal{M}$, every $x \in \mathcal{X}$, and every $f \in \hat{\mathcal{F}}_\alpha$, $\mathbb{P}[\mathsf{Detect}(\kappa, x, \tilde{y}) = \mathtt{false} \wedge \tilde{y} \neq y : (\kappa, \mathsf{Model}_\kappa) \leftarrow_\$ \mathsf{Watermark}(1^\lambda, \mathsf{Model}), \ y \leftarrow_\$ \mathsf{Model}_\kappa(x); \ \tilde{y} \leftarrow_\$ f(y)] \leq \mathsf{negl}(\lambda)$.

In our results, we primarily consider small constants $\alpha > 0$. When we refer to $\hat{\mathcal{F}}_\alpha^{\mathrm{ind}}$, the impossibility results become even stronger: if robustness fails against independent (random, uncoordinated) edits, it necessarily fails against arbitrary (coordinated) ones.

We always require $|f(y)| = |y|$, meaning all tampering functions are length-preserving. This restriction is standard in coding theory and only strengthens our impossibility results: if robust watermarking is impossible even when the adversary must preserve output length, it is certainly impossible when more general modifications, such as insertions or deletions, are permitted.

Our model further allows the watermarking algorithm to access the generative model's internals, and allows the detector to depend on both the prompt and output. Impossibility under these permissive conditions immediately implies impossibility in any more restricted setting.

## 3  MESSAGELESS SECRET-KEY CODES

In this section, we develop a central ingredient for our theory: secret-key codes that do not encode information, but instead provide a way to test the validity of a codeword and to detect tampering. These "messageless" codes distill the core cryptographic challenges of watermarking into a simple form. We formalize their properties, such as correctness, soundness, tamper detection, and pseudo-randomness, and see how these definitions serve as the foundation for our results on watermarking.

A *messageless* secret-key codes permits to enables to (i) distinguishing valid codewords from invalid ones, and (ii) detecting whether a codeword has been tampered with (possibly maliciously) during transmission. We consider codes in the secret-key setting, where both encoding and decoding use the same secret key. This secret-key variant is also simpler to construct than public-key analogues.

Formally, a messageless secret-key code is specified by a triple of polynomial-time algorithms $\Gamma = (\mathsf{KGen}, \mathsf{Enc}, \mathsf{Dec})$ over an alphabet $\Sigma$ and a codeword space $\Sigma^n$:

$\mathsf{KGen}(1^\lambda)$**:** On input the security parameter $1^\lambda$, the randomized key-generation algorithm outputs a secret key $sk$.

$\mathsf{Enc}(sk)$**:** On input $sk$, the (possibly randomized) encoding algorithm outputs a codeword $\gamma$.

$\mathsf{Dec}(sk, \gamma)$**:** On input the secret key $sk$ and a codeword $\gamma$, the deterministic decoding algorithm outputs a symbol $y \in \{\mathtt{valid}, \mathtt{invalid}, \mathtt{tampered}\}$.

A messageless secret-key code $\Gamma$ should satisfy correctness: an honestly generated codeword is always classified as $\mathtt{valid}$ by the decoding procedure. We provide the formal definition in Appendix A.4. For security, we require two properties. The first is soundness, which means that decod-

ing any fixed string (not produced by the encoder) yields `invalid` except with negligible probability over the choice of the secret key $sk$. In other words, the code identifies codewords associated to $sk$.

**Definition 4** (Soundness of messageless secret-key codes). We say that a messageless secret-key code $\Gamma$ satisfies soundness if for every fixed codeword $\hat{\gamma} \in \Sigma^n$, $\mathbb{P}[\mathsf{Dec}(sk, \hat{\gamma}) \neq \texttt{invalid} : sk \leftarrow_\$ \mathsf{KGen}(1^\lambda)] \leq \mathsf{negl}(\lambda)$.

The second property is tamper detection with respect to a family $\mathcal{F} = \{f : \Sigma^n \to \Sigma^n\}$ of codeword modification functions (possibly randomized). This property requires that decoding a tampered codeword $\tilde{\gamma} \neq \gamma$ (with $\gamma \leftarrow_\$ \mathsf{Enc}(sk)$ and $\tilde{\gamma} \leftarrow_\$ f(\gamma)$) yields `tampered` except with negligible probability.

**Definition 5** ($\mathcal{F}$-Tamper detection of messageless secret-key codes). We say a messageless secret-key code $\Gamma$ satisfies $\mathcal{F}$-tamper detection if for any $f \in \mathcal{F}$, $\mathbb{P}[\mathsf{Dec}(sk, \tilde{\gamma}) \neq \texttt{tampered} \wedge \tilde{\gamma} \neq \gamma : sk \leftarrow_\$ \mathsf{KGen}(1^\lambda); \gamma \leftarrow_\$ \mathsf{Enc}(sk); \tilde{\gamma} \leftarrow_\$ f(\gamma)] \leq \mathsf{negl}(\lambda)$.

Some settings require a third property: pseudorandomness (i.e., the codewords are indistinguishable from random). To meet space constraint, we move the formal definition to Appendix A.4.

## 3.1 Impossibility of High Tampering Rates

A central question for messageless secret-key codes is how robust they can be to tampering: if an adversary is allowed to modify a fraction of the codeword, can the code reliably detect tampering and also maintain soundness? It turns out that these two properties are fundamentally at odds if the tampering is sufficiently strong.

To see the tension, consider the following intuition. If the set of possible tampering functions includes all constant functions (that is, for every string $\hat{\gamma}$, there is a tampering function that always outputs $\hat{\gamma}$), then the decoder faces a contradiction: soundness requires that, for almost every $\hat{\gamma}$, the decoder outputs `invalid`; yet tamper detection requires that, for any $\hat{\gamma}$ produced by tampering with a valid codeword, the decoder outputs `tampered`. Since the decoder must assign a single label to each string, it cannot satisfy both properties for the same input $\hat{\gamma}$.

This intuition extends to the case where the adversary is limited to changing only a fraction $\alpha$ of the codeword, even if changes are made independently at random across positions. We now make this precise.

Let $\Sigma$ be an alphabet of size $q > 1$, and let $n$ denote the codeword length. Define $\mathcal{F}_{\alpha n}^{\mathrm{ind}}$ as the family of functions $f$ where, for each input $\gamma \in \Sigma^n$, $f(\gamma)$ is produced by independently changing each symbol: for each position, the symbol is replaced with a new random symbol from $\Sigma$ with probability $p = 1 - 1/q$, and remains unchanged with probability $1/q$. The expected fraction of changed positions is exactly $1 - 1/q$.

Below we establish the formal theorem whose proof appears in Appendix A.9.1.

**Theorem 2.** *Let $\Gamma$ be any messageless secret-key code over $\Sigma$ with codeword length $n \geq 1$ and $|\Sigma| = q > 1$. Suppose $\Gamma$ satisfies tamper detection for the family $\mathcal{F}_{\alpha n}^{\mathrm{ind}}$ with $\alpha = (1 - 1/q)(1 + \delta)$ for some $\delta \in (0, 1)$. Then $\Gamma$ cannot satisfy soundness.*

This bound is tight for the family of independent symbol tampering functions, and matches intuition from coding theory: if too large a fraction of the codeword can be arbitrarily changed, no procedure can reliably distinguish random tampering from invalid codewords. In the case of binary codewords (that is, $\Sigma = \{0, 1\}$), the impossibility threshold becomes particularly simple:

**Corollary 1.** *Let $\Gamma$ be any messageless secret-key code with alphabet $\{0, 1\}$ and codeword space $\{0, 1\}^n$ for $n \geq 1$. If $\Gamma$ satisfies $\mathcal{F}_{\alpha n}^{\mathrm{ind}}$-tamper detection for $\alpha = (1 + \delta)/2$ for any $\delta \in (0, 1)$, then $\Gamma$ cannot also satisfy soundness.*

These impossibility results do not depend on whether the decoder ever accepts any honestly generated codeword as valid. For instance, a code could be designed so that the decoder never outputs `valid` at all; it would trivially satisfy soundness (since random strings are always labeled invalid) and could still claim to detect tampering. Our theorems show that, even allowing for such "incorrect" codes, soundness and tamper detection cannot both be satisfied at high tampering rates. This is why correctness is not assumed or required for our lower bounds.

**Remark 1.** It's worth noting that the limitations described above are even more robust than they may first appear. First, suppose we relax our expectations for soundness, allowing the decoder to make mistakes on a constant fraction $\epsilon$ of random inputs, rather than insisting on negligible error. The underlying tension remains: once the adversary is able to tamper with a sufficiently large fraction of the codeword, the code can only be sound for a fraction $\epsilon$ of random strings if it tolerates a correspondingly large tamper-detection error—specifically, at least $(1-\epsilon)\left(1-\exp\left(-\frac{\delta^2 n}{3}(1-1/q)\right)\right)$ for tampering rates $\alpha = (1 - 1/q)(1 + \delta)$ and any $\delta \in (0, 1)$. In essence, even a "lenient" code, willing to make plenty of mistakes, still cannot robustly detect high-rate tampering.

Second, these limits do not depend on the codeword length being fixed in advance. If codewords are allowed to be any length, the adversary can simply determine the length and tamper independently with the appropriate fraction of positions, and the same contradiction arises. The impossibility persists no matter how the codeword length is chosen.

Taken together, these points highlight a fundamental barrier that goes beyond the classical bounds from error-correcting codes. Here, the impossibility is a direct consequence of trying to satisfy both soundness and strong tamper detection, even in the highly flexible messageless setting.

**Supporting messages, error correction, and tight construction.** In Appendix A.5, we show that the bound on $\alpha$ of Theorem 2 also applies to secret-key codes with messages and to secret-key error-correcting codes. Moreover, in Appendix A.7, we present a simple and tight information-theoretic construction that matches the bound established in Theorem 2.

### 3.2 CONNECTING WATERMARKING AND TAMPER DETECTION

The fundamental limits we have established for messageless secret-key codes also govern the robustness of watermarking schemes for generative models. To make this connection precise, we show how any watermarking scheme that is sound and robust against tampering can be used to build a secret-key code with exactly the same security guarantees.

The intuition is straightforward: if a generative model can reliably embed a watermark that survives tampering, then by fixing a prompt and interpreting the model's outputs as codewords, we obtain a code that is robust to exactly the same set of manipulations. The code's decoder simply runs the watermark detector: if it finds a watermark, it signals tampering; otherwise, it outputs invalid.

We state this reduction for the most general case, allowing for models whose outputs can have variable length. The resulting code inherits the tamper-resilience properties of the watermarking scheme for any family of tampering functions that operates on variable-length strings. If the model always produces fixed-length outputs, the same logic applies for fixed-length tampering. Below we provide the formal theorem whose proof appears in Appendix A.9.2.

**Theorem 3.** *Assume there exists a watermarking scheme $\Pi$ for a class of generative models $\mathcal{M} = \{\mathsf{Model} : \Sigma^* \to \Sigma^*\}$ satisfying soundness and $\hat{\mathcal{F}}$-robustness (for some family $\hat{\mathcal{F}}$).[1] Then, there exists a messageless secret-key code $\Gamma$ with alphabet space $\Sigma$ and codeword space $\Sigma^*$ satisfying soundness and $\hat{\mathcal{F}}$-tamper detection.*

When the output of the generative model is a binary string (i.e., $\Sigma = \{0, 1\}$), the connection above becomes especially powerful. By mapping any robust watermarking scheme to a messageless secret-key code, we can directly apply our main impossibility result for codes to watermarking itself.

Specifically, our earlier theorem (Theorem 2) shows that no messageless secret-key code can be sound and robust to tampering beyond a certain threshold—even when the codewords may have variable length. By combining this with the reduction above, we obtain an immediate impossibility for robust watermarking of binary outputs:

**Corollary 2.** *Let $\mathcal{M} = \{\mathsf{Model} : \mathcal{X} \to \{0, 1\}^*\}$ be any class of generative models with binary outputs. There is no watermarking scheme for $\mathcal{M}$ that achieves both soundness and $\hat{\mathcal{F}}_\alpha^{\mathsf{ind}}$-robustness for any tampering rate $\alpha \geq (1 + \delta)/2$, for any $\delta \in (0, 1)$. This holds even if the model's prompt space $\mathcal{X}$ is arbitrary, and even if the soundness is relaxed to allow a constant error probability.*

---

[1]For the sake of clarity, we assume the prompt and output space of $\mathsf{Model}$ is defined over the same alphabet. The result generalizes to different alphabets as well.

The impossibility threshold we have identified for tamper-resilient codes is not just a technical detail; it serves as a universal limit for watermarking. Any watermarking scheme for generative models with binary outputs must face this same upper bound, no matter how the scheme is constructed or how input and output lengths are chosen. The recent work of Christ & Gunn (2024) provides a compelling example: their PRC-based watermarking schemes achieve robustness against independent bit-flip tampering up to the threshold $\alpha < (1 - \delta)/2$, and their approach even accommodates certain types of deletions. Our corollary shows that this rate is not only achievable but also optimal. Although we do not claim a perfect correspondence between all watermarking schemes and all PRC constructions, since this depends on the precise model and tampering family, the upper bound on robustness holds for all such schemes.

## 4  A CONCRETE ATTACK

Our goal is to test whether the theoretical robustness threshold we proved manifests in practice. For this purpose, we analyze the PRC watermark of Gunn, Zhao, and Song (Gunn et al., 2025), implemented in Stable Diffusion 2.1 Base (512×512 resolution, 50 denoising steps). This watermark is representative of the state of the art: it preserves perceptual quality, is provably undetectable under standard assumptions, and resists a wide class of natural manipulations.

The scheme embeds a PRC (pseudorandom) codeword in the latent space. Let $\mathbf{v} \in \mathbb{R}^{4 \times 64 \times 64}$ denote the latent tensor produced by the diffusion model. For each entry $v_i$ of $\mathbf{v}$, the sign is aligned with the corresponding codeword bit $\gamma_i \in \{0, 1\}$ where $\gamma = \gamma_1 || \ldots || \gamma_n$ is the bit composition of a PRC codeword $\gamma$. If $\gamma_i = 1$, the sign is left unchanged; if $\gamma_i = 0$, the sign is flipped. The magnitudes of the latent remain untouched, so the watermarked latent still follows the Gaussian distribution required for high-quality image generation. The codeword is pseudorandom, so without the detection key the modified latent is indistinguishable from a fresh Gaussian sample.

Detection proceeds by first inverting a generated image $\hat{x}$ back into a latent $\hat{\mathbf{v}}$ using an approximate inversion procedure, and then comparing $\text{sign}(\hat{\mathbf{v}})$ to the expected PRC codeword $\gamma$. Because inversion is lossy, $\hat{\mathbf{v}}$ typically agrees with the original latent on about 90% of entries even in the absence of any edits. To account for this, the detector employs a belief-propagation (BP) decoder that attempts to correct errors and recover the original codeword. The key parameter is the pre-decoding sign error rate. As reported in Gunn et al. (2025), the BP decoder is probabilistic and exhibits variable robustness: it sometimes recovers the codeword under substantial perturbations, but overall it is less reliable than the detector, which remains robust across a wide range of attacks. In our experiments we observe that once the pre-decoding error rate approaches one-half of the latent positions, the BP decoder fails consistently.

This threshold provides a benchmark for our study. The central question is whether simple image manipulations can drive the recovered latent to the 50% error mark, thereby erasing the watermark.

### 4.1  FIRST ATTEMPTS

Before diving into our attack, we tested the robustness of Gunn et al. (2025) (while considering quality preservation), by choosing a broad set of common image manipulations (including gaussian noise, gaussian blur, pixel-wise color shifts, lossy format conversions). To meet space constraints, the results of our experiments are presented in Appendix A.8. They demonstrate that, although the images were visibly altered (sometimes severely), yet the watermark survived. From this outcomes, we can draw a clear lesson: edits that preserve image quality produce far too few sign flips to reach the 50% threshold, while edits that come closer to the threshold do so only at the cost of destroying the image. To break the watermark without sacrificing fidelity, a qualitatively different kind of transformation is required.

### 4.2  OUR SUCCESSFUL ATTACK

The failure of local manipulations suggests what is missing. Noise, blur, color shifts, and compression alter pixel values but preserve the coordinate system of the image. The encoder still sees essentially the same structure, and so the latent sign pattern (i.e., the PRC codeword $\gamma$) drifts only modestly. To break the watermark, we sought an operation that would force the encoder to reinter-

pret the image globally (significantly impacting the encoded codeword $\gamma$), while leaving the picture perceptually unchanged.

This reasoning led to the crop-and-resize attack. We take a watermarked $512 \times 512$ image, crop 15 pixels from each side to obtain a $482 \times 482$ image, and then resize it back to $512 \times 512$ using bicubic interpolation.[2] The crop removes only a narrow border, and the interpolation smoothly reconstructs missing pixels from their neighbors. Visually, the result is nearly indistinguishable from the original. Examples are shown in Figure 1, where side-by-side comparisons reveal no perceptible difference between the original and the cropped-and-resized images.

The effect on the watermark is decisive. Across more than one thousand test cases, the recovered latent after crop-and-resize showed pre-decoding error rates concentrated near $50\%$. This error rate is exactly the threshold at which BP decoding fails. Indeed, in every trial the decoder was unable to recover the codeword, and detection failed completely. Unlike the earlier attacks, which either left the watermark intact or visibly degraded the image, this single transformation erased the watermark while keeping the content unchanged.

In short, crop-and-resize does what no other edit could: it simultaneously preserves perceptual quality and drives the latent representation to the theoretical boundary. The watermark disappears, not because the image is damaged, but because the latent has been resampled in a way that overwhelms the error-correcting capacity of the code.

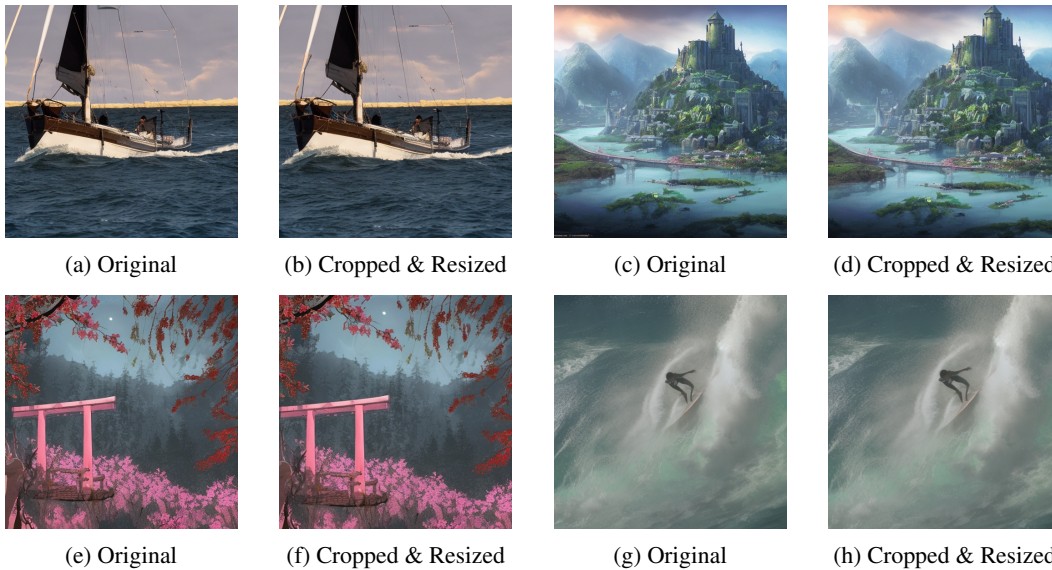

(a) Original     (b) Cropped & Resized     (c) Original     (d) Cropped & Resized

(e) Original     (f) Cropped & Resized     (g) Original     (h) Cropped & Resized

Figure 1: Comparison between original and crop & resize attacked images.

**Why It Works.** The success of crop-and-resize can be traced to how the encoder interprets the image. Local edits (noise, blur, color shifts, compression) perturb pixel values but leave the grid of the image intact. The encoder still recognizes the same arrangement, and the latent sign pattern changes only in scattered places. Cropping followed by resizing is different. Cropping removes a thin border, and resizing does not simply stretch the remaining pixels. It recomputes every pixel value by bicubic interpolation on a new lattice. The encoder is then asked to describe the same visual content in a new coordinate system. This global resampling is what drives nearly half of the latent entries across zero, flipping their signs.

**Latent-space effects.** Figure 2 shows this effect in detail. The original latent sign pattern (top left) encodes the watermark (i.e., the PRC codeword) as a pseudorandom arrangement of signs. After crop-and-resize, the recovered latent (bottom left) is visibly scrambled. The difference map before decoding (top right) shows that $48.07\%$ of the signs have flipped. This is essentially the

---

[2] We use a $15\,\mathrm{px}$ crop by default; for tightly framed images a smaller crop (e.g., $10\,\mathrm{px}$) suffices, as in Figure 6.

information-theoretic boundary: once errors approach $50\%$, the codeword is indistinguishable from random. This is exactly the core argument to prove the impossibility result (proven by demonstrating a generic attack) of Theorem 2. The bottom-right panel shows the result after belief propagation, the error-correcting decoder, has attempted to repair the errors. At this level of corruption, BP cannot recover the original codeword and instead converges to a different pseudorandom codeword. As a result, the measured error rises slightly to $48.96\%$. This increase is not paradoxical but diagnostic: the decoder has lost all correlation with the true codeword. The image itself is unchanged, but the watermark is irretrievable.

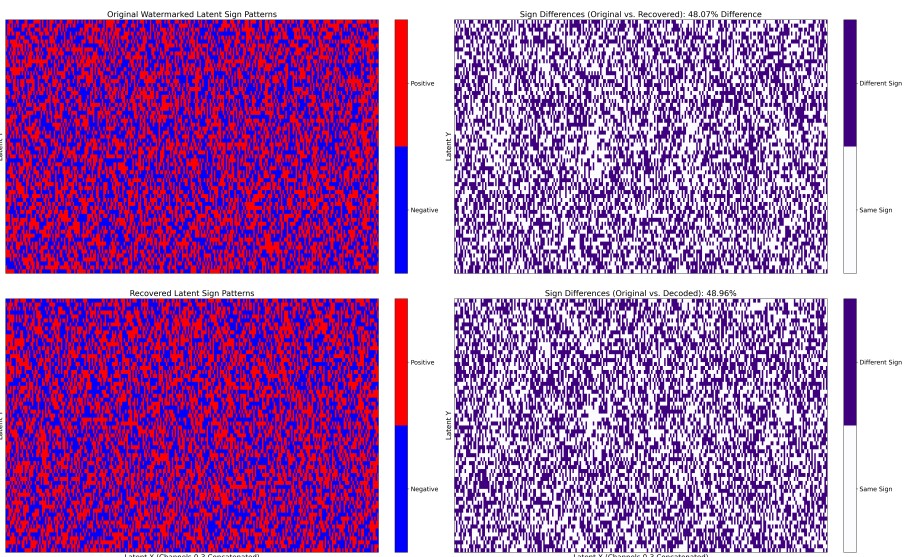

Figure 2: Latent sign analysis of a watermarked image before and after the crop and resize attack. Top left: the original latent sign pattern (red = positive, blue = negative) that encodes the watermark as a pseudorandom codeword. Bottom left: the latent recovered from the attacked image, which is visibly scrambled compared to the original. Top right: a difference map comparing the original and recovered latents, showing that $48.07\%$ of signs have flipped (purple = flipped, white = unchanged). Bottom right: a difference map after belief propagation, the error-correcting decoder, has tried to repair the errors. Because the error rate is already near the $50\%$ threshold, the decoder fails and converges to a different pseudorandom codeword. The resulting error rate increases slightly to $48.96\%$, confirming that the watermark cannot be recovered.

**Variants.** Other global manipulations tell the same story. Figure 4 compares crop and resize with several related operations. Cropping and padding with black pixels produced about $16.7\%$ error and left a conspicuous border. Downscaling and then upscaling, even aggressively to $312 \times 312$ before resizing back, produced about $12.1\%$ error. Neither came close to the threshold. Downscaling and padding with black pixels did reach $\sim 50\%$ error (Figure 5), but the added border made the alteration visually obvious, as shown in subfigure (d) of Figure 4. In contrast, crop and resize combined two properties: it perturbed the latent enough to destroy the watermark, and it did so while preserving the appearance of the image.

**Generalization.** The vulnerability is not limited to this specific setup. Generative models often allow users to control framing explicitly. An adversary can request an image with an artificial border and then remove it by cropping and resizing. Figure 6 illustrates this approach: the model produces an image with a frame, the border is removed, and the watermark disappears while the content remains intact. No knowledge of the watermark key is needed, and no optimization is required. The principle is general: any transformation that globally resamples the image while keeping its perceptual content unchanged will drive the latent to the robustness threshold.

## ETHICS STATEMENT

Our study concerns the limits of watermarking schemes for generative models. It does not involve human subjects, personal data, or datasets with ethical concerns. The results highlight that certain vulnerabilities are not accidental but inherent: they arise from the structure of watermarking itself and can be exploited regardless of our work. Making these limitations explicit serves the goal of building more reliable methods, by clarifying where robustness is and is not possible. We therefore view the main effect of this paper as reducing long-term risk rather than creating new avenues for misuse. The work has no proprietary or commercial ties and does not differentially affect particular groups.

## REPRODUCIBILITY STATEMENT

We have taken the following steps to ensure the reproducibility of our results. For the theoretical contributions, the appendix provides complete proofs of all statements, theorems, and corollaries. For the practical contributions, the source code for reproducing all experiments, including the implementation of the proposed attack, is available in the following anonymous repository: https://anonymous.4open.science/r/PRC-Attacker-2B6E/. The repository also contains detailed instructions for executing the code.

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

# A APPENDIX

## A.1 RELATED WORK

Watermarking refers to the process of embedding a signal in generated content such as images (Boland et al., 1995; Cox et al., 2007; Hayes & Danezis, 2017; Ruanaidh & Pun, 1997; Zhu et al., 2018), text (Atallah et al., 2001; 2003), audio (Arnold, 2000; Boney et al., 1996), or video that is imperceptible to humans but algorithmically detectable. Its primary use is to enable attribution or provenance without significantly degrading content quality. The study of watermarking spans decades and includes both practical systems and formal analyses, with results on constructing watermarking schemes as well as breaking or proving impossibility. With the rise of modern generative models, watermarking has become a sharper test of the trade-offs between imperceptibility, robustness, and efficiency.

**Constructions.** Watermarking methods are commonly classified based on when the watermark is applied: either after generation (post-processing) or during the generation process itself (in-processing). Post-processing schemes embed the watermark into the output after it has been generated. These approaches often degrade output quality (Cox et al., 2007; Wan et al., 2022; An et al., 2024). In-processing schemes, by contrast, embed the watermark during content generation. They were first introduced in image generators (Yu et al., 2021), and later adapted to large language models (LLMs) (Kirchenbauer et al., 2023). Image watermarking techniques include modifying weights through pretraining or fine-tuning (Fei et al., 2022; Fernandez et al., 2023; Zeng et al., 2023; Zhao et al., 2023), or intervening in the sampling trajectory of diffusion models (Wen et al., 2023). In the LLM setting, most methods operate at inference time by biasing token selection in a way that encodes a hidden message (Kirchenbauer et al., 2023; Christ et al., 2024). Some schemes further ensure that the overall distribution over outputs remains statistically indistinguishable from unmarked outputs (Kuditipudi et al., 2024), preserving quality.

A more recent line of work leverages cryptographic and coding-theoretic tools to design undetectable watermarking schemes. One direction (Christ & Gunn, 2024) proceeds via pseudorandom error-correcting codes (PRCs) as the watermark signals. These codes can be constructed using low-density parity-check (LDPC) codes (Christ & Gunn, 2024; Alrabiah et al., 2025; Ghentiyala & Guruswami, 2024) and allow for robust detection under edits. PRC-based watermarking was recently applied to images (Gunn et al., 2025). The main limitation (Gunn et al., 2025) is that although the underlying PRC is robust to modifications on a bounded number of bits, the resulting PRC-based watermarking is robust only to independent changes, limiting its applicability in real-world scenarios. In parallel, Fairoze et al. (2025) propose a public watermarking scheme that uses digital signatures, allowing third parties to verify the watermark and ensuring robustness as long as a small window of the generated content remains unmodified.

**Attacks and Impossibilities.** Alongside construction efforts, recent work has explored the limitations of existing watermarking schemes. In text, the most prominent attack is paraphrasing (Cheng et al., 2025; Krishna et al., 2023; Zhang et al., 2023; 2024), rewriting the watermarked content while preserving its meaning, using language models, translation tools, or manual edits. These attacks often succeed partially, but at the cost of degraded output quality. In the vision domain, a range of attacks aim to remove watermarks by adding noise to the image or latent representation, or by using optimization techniques to remove the watermark (Lukas et al., 2024; Saberi et al., 2024; Zhao et al., 2024). Many of these attacks have limitations. For example, Zhao et al. (2024) focus on post-hoc watermarking methods, while Saberi et al. (2024) require either white-box access to the detector or access to many watermarked and unwatermarked examples.

A more general impossibility result by Zhang et al. (2024; 2023) shows that constructing robust watermarking schemes is impossible under certain conditions. They model generated outputs as nodes in a graph and prove that if an adversary has access to two oracles (a quality oracle that tells the adversary whether the output is still valid, and a perturbation oracle that allows small edits preserving semantics) then the adversary can walk through the graph until the watermark is erased without degrading content quality. This result does not specify an exact condition for when removal occurs, only that eventually such a walk succeeds.

In contrast, our result makes no oracle assumptions and provides a concrete lower bound. We show that if an adversary is willing to change more than half of the bits, then the watermark is surely removed. Prior work demonstrates that robust watermarking cannot be sustained indefinitely. Our result identifies the precise threshold at which robustness fails.

## A.2   NOTATION

Small letters (such as $x$) denote individual objects or values; calligraphic letters (such as $\mathcal{X}$) denote sets; sans-serif letters (such as A) denote algorithms. For a string $x \in \{0, 1\}^*$, $|x|$ denotes its length. For a set $\mathcal{X}$, $|\mathcal{X}|$ is its cardinality. If $x$ is chosen uniformly at random from $\mathcal{X}$, we write $x \leftarrow_{\$} \mathcal{X}$. The Hamming distance $\mathrm{dist}(x, x')$ between two strings $x$ and $x'$ (over an alphabet $\Sigma$) is the number of positions where they differ.

For a deterministic algorithm A, $y = \mathsf{A}(x)$ means that $y$ is the output of A on input $x$. For a randomized algorithm A, we write $y \leftarrow_{\$} \mathsf{A}(x)$ to indicate $y$ is sampled from the output distribution of A on input $x$. Alternatively, $y = \mathsf{A}(x; r)$ denotes the output when A runs on $x$ with randomness $r$. An algorithm A is called *probabilistic polynomial-time* (PPT) if it is randomized and always halts in time polynomial in $|x|$.

We use $\mathsf{negl}(\lambda)$ for an arbitrary negligible function of the security parameter $\lambda \in \mathbb{N}$, i.e., for all $c > 0$, $\mathsf{negl}(\lambda) = o(\lambda^{-c})$. We use $\mathsf{poly}(\lambda)$ to denote a polynomial function of $\lambda$. Unless noted otherwise, all algorithms receive the security parameter $1^\lambda$ as input.

## A.3   DEFINITION OF CORRECTNESS OF WATERMARKING SCHEMES

**Definition 6** (Correctness of watermarking). Let $\mathcal{M} = \{\mathsf{Model} : \mathcal{X} \to \mathcal{Y}\}$ be a class of generative models. We say that $\Pi$ satisfies *correctness* if for every $\mathsf{Model} \in \mathcal{M}$ and for every $x \in \mathcal{X}$, $\mathbb{P}\big[\mathsf{Detect}(\kappa, x, y) = \mathtt{true} : (\kappa, \mathsf{Model}_\kappa) \leftarrow_{\$} \mathsf{Watermark}(1^\lambda, \mathsf{Model}), \ y \leftarrow_{\$} \mathsf{Model}_\kappa(x)\big] = 1$.

## A.4   DEFINITIONS OF CORRECTNESS AND PSEUDORANDOMNESS OF MESSAGELESS SECRET-KEY CODES

**Definition 7** (Correctness of messageless secret-key codes). We say that a messageless secret-key code $\Gamma$ satisfies correctness if for all $\lambda \in \mathbb{N}$ and for all $sk \in \mathsf{KGen}(1^\lambda)$, it holds that $\mathsf{Dec}(sk, \mathsf{Enc}(sk)) = \mathtt{valid}$. If Enc is randomized, we require $\mathbb{P}[\mathsf{Dec}(sk, \mathsf{Enc}(sk)) = \mathtt{valid}] = 1$ where the probability is over the randomness of the encoding.

Intuitively, a messageless secret-key code is pseudorandom if the output of Enc is indistinguishable from a uniformly random string of the same length, even if Enc is called multiple times. This property requires that Enc is randomized (since a deterministic encoder is trivially distinguishable from random).

**Definition 8** (Pseudorandomness of messageless secret-key codes). We say that a messageless secret-key code $\Gamma$ satisfies pseudorandomness if for every PPT distinguisher D,

$$\big|\mathbb{P}\big[\mathsf{D}^{\mathcal{O}_{\mathsf{real}}}(1^\lambda) = 1 : sk \leftarrow_{\$} \mathsf{KGen}(1^\lambda)\big] - \mathbb{P}\big[\mathsf{D}^{\mathcal{O}_{\mathsf{rand}}}(1^\lambda) = 1\big]\big| \le \mathsf{negl}(\lambda),$$

where the inputless oracle $\mathcal{O}_{\mathsf{real}}$ (resp. $\mathcal{O}_{\mathsf{rand}}$), each time it is invoked, returns $\mathsf{Enc}(sk)$ (resp. $\rho \leftarrow_{\$} \Sigma^n$).

## A.5   SUPPORTING MESSAGES AND ERROR CORRECTING SECRET-KEY CODES

Just as in the messageless case, we can define secret-key codes that are designed not only to detect tampering, but also to carry and recover messages. Here, the code serves two roles at once: if a codeword has not been altered, it should decode back to the original message; if the codeword has been tampered with or is otherwise invalid, the decoder should indicate this fact.

Formally, such a code consists of three algorithms: a key generator KGen, an encoder Enc that takes both a secret key and a message and outputs a codeword, and a decoder Dec that uses the secret key to recover either the original message, or a special symbol (`invalid` or `tampered`) indicating something has gone wrong.

As before, we will be interested in the fundamental properties of such codes: correctness, soundness, tamper detection, and pseudorandomness. The presence of a message simply means that correctness now asks for exact message recovery from an unaltered codeword, and the other properties extend in a natural way. For completeness, we give their formal definitions below.

Formally, a *secret-key multi-message code* is defined by a triple of polynomial-time algorithms $\Gamma = (\mathsf{KGen}, \mathsf{Enc}, \mathsf{Dec})$, specified as follows:

$\mathsf{KGen}(1^\lambda)$**:** A randomized key-generation algorithm that, on input the security parameter, outputs a secret key $sk$.

$\mathsf{Enc}(sk, \mu)$**:** A (possibly randomized) encoding algorithm that, given the secret key $sk$ and a message $\mu \in \Sigma^m$, outputs a codeword $\gamma \in \Sigma^n$.

$\mathsf{Dec}(sk, \gamma)$**:** A deterministic decoding algorithm that, given the secret key $sk$ and a codeword $\gamma$, outputs either a message in $\Sigma^m$, or one of the symbols `invalid` or `tampered`.

The key properties for such codes are defined as follows:

**Definition 9** (Correctness)**.** A secret-key multi-message code $\Gamma$ satisfies *correctness* if for all $\lambda \in \mathbb{N}$, all $sk \in \mathsf{KGen}(1^\lambda)$, and all messages $\mu \in \Sigma^m$, $\mathbb{P}[\mathsf{Dec}(sk, \mathsf{Enc}(sk, \mu)) = \mu] = 1$, where the probability is taken over the randomness of the encoding, if any.

**Definition 10** (Soundness)**.** A secret-key multi-message code $\Gamma$ satisfies *soundness* if for every fixed codeword $\hat{\gamma} \in \Sigma^n$, $\mathbb{P}[\mathsf{Dec}(sk, \hat{\gamma}) \neq \texttt{invalid} : sk \leftarrow_\$ \mathsf{KGen}(1^\lambda)] \leq \mathsf{negl}(\lambda)$.

**Definition 11** ($\mathcal{F}$-Tamper Detection)**.** $\Gamma$ satisfies $\mathcal{F}$-*tamper detection* if for any tampering function $f \in \mathcal{F}$ and any message $\mu \in \Sigma^m$,

$$\mathbb{P}[\mathsf{Dec}(sk, \tilde{\gamma}) \neq \texttt{tampered} \land \tilde{\gamma} \neq \gamma : sk \leftarrow_\$ \mathsf{KGen}(1^\lambda), \ \gamma \leftarrow_\$ \mathsf{Enc}(sk, \mu), \ \tilde{\gamma} = f(\gamma)] \leq \mathsf{negl}(\lambda).$$

**Definition 12** (Pseudorandomness)**.** $\Gamma$ satisfies *pseudorandomness* if for every PPT distinguisher D, $\left| \mathbb{P}[\mathsf{D}^{\mathcal{O}_{\mathsf{real}}}(1^\lambda) = 1 : sk \leftarrow_\$ \mathsf{KGen}(1^\lambda)] - \mathbb{P}[\mathsf{D}^{\mathcal{O}_{\mathsf{rand}}}(1^\lambda) = 1] \right| \leq \mathsf{negl}(\lambda)$, where $\mathcal{O}_{\mathsf{real}}(\mu)$ returns $\mathsf{Enc}(sk, \mu)$ and $\mathcal{O}_{\mathsf{rand}}(\mu)$ returns a uniformly random string from $\Sigma^n$.

**Remark 2.** Our impossibility results extend from messageless codes to codes that also carry messages. The reason is simple: if you have a code that can both recover messages and detect tampering, you can always just ignore the message part and use the code to check whether an input is valid, tampered, or invalid. In other words, any code that solves the harder message-recovery problem automatically solves the simpler messageless problem as well. So if it is impossible to achieve soundness and tamper detection beyond a certain threshold without messages, it is also impossible when messages are present.

## A.6 FROM TAMPER DETECTION TO ERROR CORRECTION

The barrier we have identified for tamper detection in messageless codes also holds in the stronger setting of error correction for codes that encode messages. This is because error correction necessarily includes the task of detecting whether a codeword has been tampered with: if a code can recover the message, it can certainly tell when decoding fails.

Formally, a code achieves error correction for a family of tampering functions $\mathcal{F}$ if, whenever a codeword $\gamma$ encoding a message $\mu$ is tampered with to produce $\tilde{\gamma} \neq \gamma$, the decoder outputs the original message $\mu$ except with negligible probability:

**Definition 13** ($\mathcal{F}$-Error Correction)**.** A secret-key multi-message code $\Gamma$ achieves $\mathcal{F}$-error correction if, for all $f \in \mathcal{F}$ and all $\mu$, when $sk \leftarrow_\$ \mathsf{KGen}(1^\lambda)$, $\gamma \leftarrow_\$ \mathsf{Enc}(sk, \mu)$, and $\tilde{\gamma} = f(\gamma)$ with $\tilde{\gamma} \neq \gamma$, we have $\mathbb{P}[\mathsf{Dec}(sk, \tilde{\gamma}) \neq \mu] \leq \mathsf{negl}(\lambda)$.

A code with correctness, soundness, error correction, and pseudorandomness is a pseudorandom error-correcting code (PRC) (Christ & Gunn, 2024). In our model, the decoder may output a tampered flag, but this does not affect the fundamental limits.

The key point is that impossibility for messageless codes immediately implies impossibility for message-carrying codes. If a code can correct errors for even a single message, it can be viewed as a messageless code by fixing that message and treating every successful decoding as "valid." The converse also holds. To obtain a multi-message code from a single-message code, one can apply the

standard transformation of Christ & Gunn (2024), which converts any single-message PRC into a multi-message one, preserving soundness and error correction.

**Theorem 4** (Extension of Impossibility). *Let $\mathcal{F}$ be any family of tampering functions. If no messageless secret-key code achieves soundness and $\mathcal{F}$-tamper detection above a given tampering rate, then no secret-key multi-message code (including PRCs) achieves error correction for messages above that rate.*

*Proof.* Suppose, for contradiction, that there exists a secret-key multi-message code $\Gamma$ achieving $\mathcal{F}$-error correction above the threshold. Fix any message $\mu^*$. Define a messageless code $\Gamma'$ by $\Gamma'.\mathsf{Enc}(sk) = \Gamma.\mathsf{Enc}(sk, \mu^*)$ and $\Gamma'.\mathsf{Dec}(sk, \gamma) = \texttt{valid}$ if $\Gamma.\mathsf{Dec}(sk, \gamma) = \mu^*$, and $\texttt{invalid}$ otherwise. The error correction property for $\Gamma$ ensures tamper detection for $\Gamma'$, contradicting the impossibility for messageless codes. For the multi-message setting, the construction of Christ & Gunn (2024) can be applied to lift any single-message PRC to a multi-message PRC, preserving all security properties. $\square$

**Corollary 3.** *No secret-key code (including PRCs (Christ & Gunn, 2024)) over a binary alphabet can achieve error correction at tampering rates exceeding $(1+\delta)/2$, for any constant $\delta > 0$, without losing soundness.*

Thus, the impossibility is fundamental: it holds for all cryptographically meaningful coding schemes, regardless of whether they detect or correct errors, or how many messages they support.

### A.7 A Construction of Messageless Secret-key Codes

To demonstrate that the tightness of the impossibility threshold for robust watermarking, we give a simple information-theoretic construction of a messageless secret-key code that attains optimal soundness and tamper detection up to the threshold. This construction is explicit and efficient, with all operations linear in the codeword length.

The main idea is to use the secret key itself as the codeword, and have the decoder distinguish between "valid," "tampered," and "invalid" based on Hamming distance from the key. This divides the space of possible codewords into three regions, separated by a threshold $t$.

We now formalize this construction.

**Construction 1.** Let $n$ be the codeword length, and fix a parameter $\delta \in (0, 1)$. Define $t = n(1 - \frac{1}{q})(1 - \delta)$. The code $\Gamma = (\mathsf{KGen}, \mathsf{Enc}, \mathsf{Dec})$ is defined as:

**Key Generation** $\mathsf{KGen}(1^\lambda)$**:** Sample a secret key $sk \leftarrow_\$ \Sigma^n$ uniformly at random, where $|\Sigma| = q = n^c$ for constant $c > 1$.

**Encoding** $\mathsf{Enc}(sk)$**:** Output the codeword $\gamma = sk$.

**Decoding** $\mathsf{Dec}(sk, \gamma)$**:** Given $\gamma \in \Sigma^n$:

- If $\text{dist}(\gamma, sk) > t$, output $\texttt{invalid}$.
- If $0 < \text{dist}(\gamma, sk) \leq t$, output $\texttt{tampered}$.
- If $\gamma = sk$, output $\texttt{valid}$.

The correctness property for this construction is immediate: the decoder always accepts the honest codeword, which is just the secret key.

What remains are the two core security properties: soundness, and robust detection of tampering up to the optimal threshold. Both follow from a simple analysis of Hamming distance and concentration.

**Theorem 5** (Security of the Simple Construction). *Let $n$ and $q = |\Sigma|$ such that $\frac{n}{q} \in \omega(\log \lambda)$.[3] The messageless secret-key code $\Gamma$ from Construction 1 satisfies soundness and $\mathcal{F}_{\alpha n}$-tamper detection for $\alpha = (1 - \frac{1}{q})(1 - \delta)$ for any $\delta \in (0, 1)$.*

---

[3]This can be always enforced by increasing the codeword length $n$ w.r.t. the cardinality $q$ of $\Sigma$.

*Proof.* We prove each property in turn.

**Soundness:** Fix any string $\hat{\gamma} \in \Sigma^n$ and random key $sk \leftarrow_\$ \Sigma^n$. Let $X$ be the number of positions where $\hat{\gamma}[i] = sk[i]$; $X$ is binomial with mean $n/q$. The decoder outputs `invalid` unless $X > n(1 + \delta)/q$ (note that the last inequality holds whenever $q \geq 2$).[4]

By Chernoff bound,

$$\mathbb{P}\left[X \geq \frac{n}{q}(1 + \delta)\right] \leq \exp\left(-\frac{\delta^2 n}{2q}\right)$$

which is negligible whenever $\frac{n}{q} \in \omega(\log \lambda)$ and constant $\delta > 0$.

**Tamper Detection:** Fix any $f \in \mathcal{F}_{\alpha n}$ with $\alpha = (1 - \frac{1}{q})(1 - \delta)$. Let $\gamma = sk$ and $\tilde{\gamma} = f(\gamma)$, with $\tilde{\gamma} \neq \gamma$. By definition of $f$, $\text{dist}(\tilde{\gamma}, sk) \leq n(1 - \frac{1}{q})(1 - \delta) = t$. Thus, the decoder outputs `tampered`.

Both properties hold as claimed. □

**Achieving Pseudorandomness.** We can upgrade any deterministic messageless secret-key code to achieve pseudorandomness as follows:

**Construction 2** (Pseudorandom Messageless Secret-Key Code with Public Counter)**.** Let $\Gamma = (\mathsf{KGen}, \mathsf{Enc}, \mathsf{Dec})$ be any messageless secret-key code with codeword length $n$. Let $\mathsf{F} : \{0,1\}^\lambda \times \{0,1\}^p \rightarrow \{0,1\}^{n \log q}$ be a secure pseudorandom function. Define the upgraded code $\Gamma' = (\mathsf{KGen}', \mathsf{Enc}', \mathsf{Dec}')$:

**Key Generation** $\mathsf{KGen}'(\lambda)$**:** Sample $(sk, \kappa) \leftarrow_\$ \Gamma.\mathsf{KGen}(\lambda) \times \{0,1\}^\lambda$.

**Encoding** $\mathsf{Enc}'((sk, \kappa), \pi)$**:** Output $\gamma' = \Gamma.\mathsf{Enc}(sk) \oplus \mathsf{F}(\kappa, \pi)$, where $\pi \in \{0,1\}^p$ is a public counter, where $\Gamma.\mathsf{Enc}(sk)$ is binary.

**Decoding** $\mathsf{Dec}'((sk, \kappa), \gamma', \pi)$**:** Compute $\gamma = \gamma' \oplus \mathsf{F}(\kappa, \pi)$ and output $\Gamma.\mathsf{Dec}(sk, \gamma)$.

**Requirements on the Counter.** To claim security, the value $\pi$ must satisfy the following three conditions: $(i)$ Publicly known to both encoder and decoder, $(ii)$ never reused (each $\pi$ is used at most once), and $(iii)$ not settable or rewritable by the adversary.

Any deterministic, sound, and tamper-detecting code $\Gamma$ yields a pseudorandom code $\Gamma'$ as above, provided a secure PRF and a suitable counter implementation. In practice, such a counter may be realized as a blockchain index, a hardware monotonic counter, or a database sequence number, as long as it is monotonic and trusted. All security properties are preserved, and codewords are now pseudorandom to any party not holding $\kappa$.

## A.8 ROBUSTNESS OF GUNN ET AL. (2025) AGAINST COMMON IMAGE MANIPULATIONS ATTACK VECTORS

**Image manipulations.** Gaussian noise, Gaussian blur, HSV perturbations, and attribute adjustments (saturation, hue, exposure, contrast) were applied to watermarked images. Even when these visibly degraded the images, the pre-decoding error rate never exceeded 26%, and BP reduced the error to below 1%. The watermark remained detectable in every case.

**Pixel-wise color shifts.** Next, we modified each pixel by adding fixed or randomized RGB offsets. A uniform shift of $(10, 0, 0)$, making the image slightly redder, produced under 10% error. A large uniform shift of $(75, 75, 75)$, which washed out the image, raised the error to only 12%. Randomized RGB perturbations across all pixels, with values drawn uniformly between 0 and 50, produced slightly higher disruption but never more than 23%. These results suggest that uniform color manipulations do not significantly disturb the sign pattern in latent space.

---

[4] Indeed, since $X$ is the random variable representing the number of identical symbols, the number of differing symbols is $n - X \leq n - n(1 + \delta)/q = n\left(1 - \frac{1}{q}\right)(1 + \delta) = t$, which is exactly the condition under which the decoder outputs either `tampered` or `valid`.

**Lossy format conversions.**   Finally, we tested whether re-encoding the image in lossy formats could erase the watermark. JPEG compression at 15% quality produced heavy visible degradation and a 32% pre-decoding error rate, which BP reduced to 10%. WebP conversion at similar quality gave 34% error, reduced to 15% after decoding. In both cases, the watermark remained detectable.

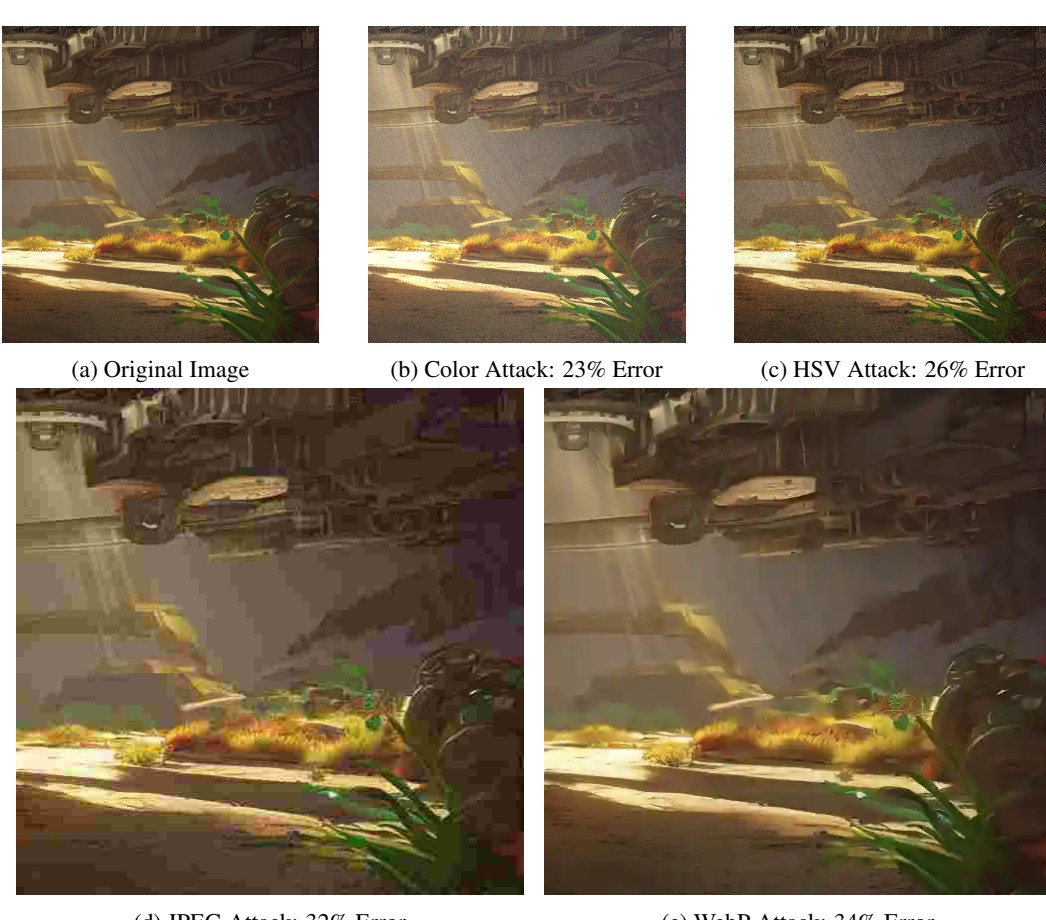

(a) Original Image          (b) Color Attack: 23% Error          (c) HSV Attack: 26% Error

(d) JPEG Attack: 32% Error                    (e) WebP Attack: 34% Error

Figure 3: A series of images showing various attacks on a watermarked image, none of which removes the watermark

Representative examples of these attacks are shown in Figure 3. The images have been visibly altered (sometimes severely) yet the watermark survived. The lesson is clear: edits that preserve image quality produce far too few sign flips to reach the 50% threshold, while edits that come closer to the threshold do so only at the cost of destroying the image. To break the watermark without sacrificing fidelity, a qualitatively different kind of transformation is required.

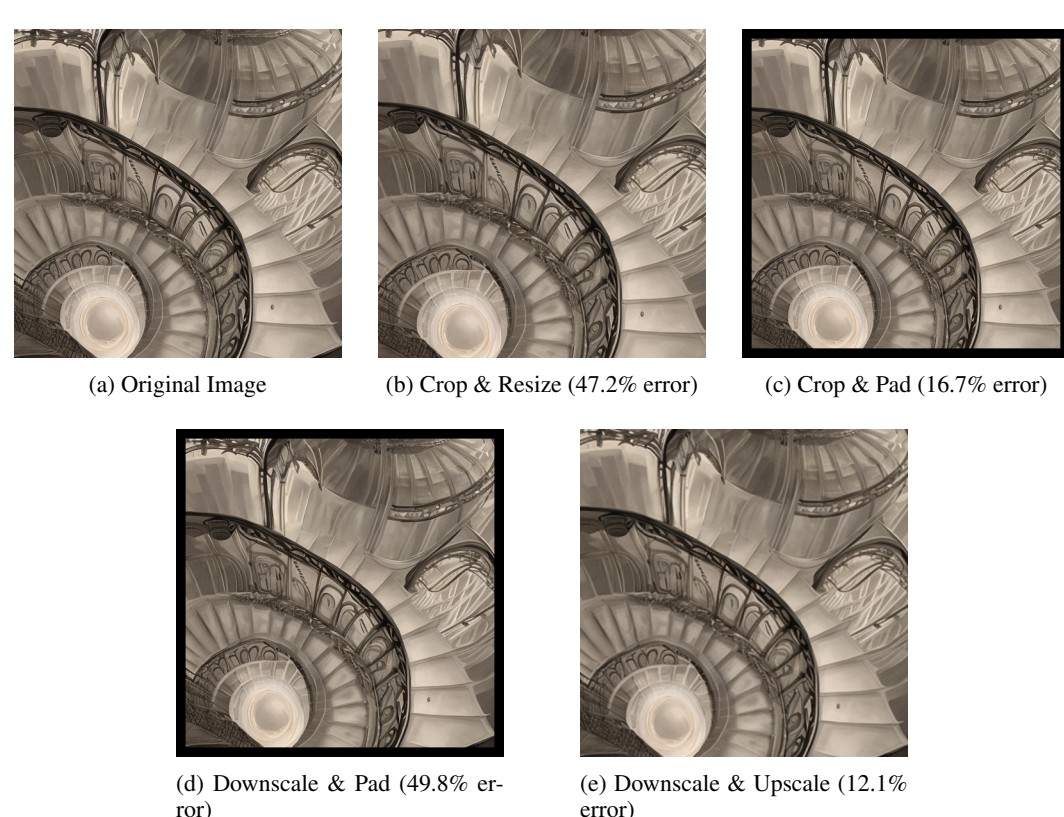

(a) Original Image     (b) Crop & Resize (47.2% error)     (c) Crop & Pad (16.7% error)

(d) Downscale & Pad (49.8% error)     (e) Downscale & Upscale (12.1% error)

Figure 4: Visualization of this branch of attacks (b-e) compared to the original (a)

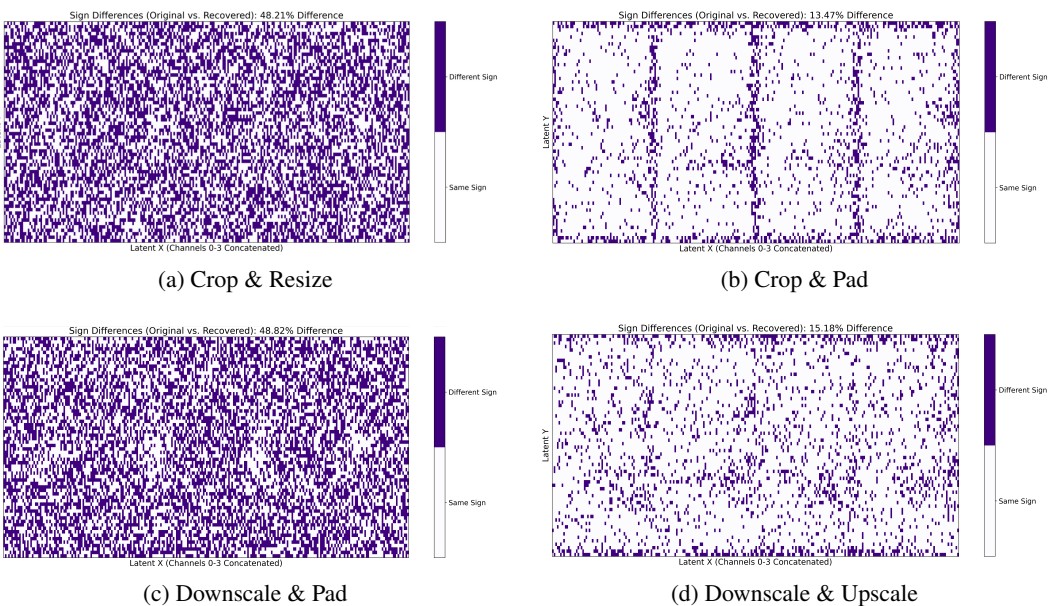

(a) Crop & Resize     (b) Crop & Pad

(c) Downscale & Pad     (d) Downscale & Upscale

Figure 5: Side-by-side comparison of latent space visualizations for the main attack and each variant

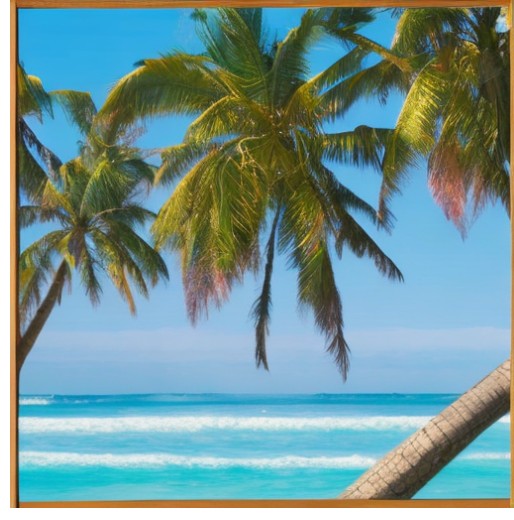 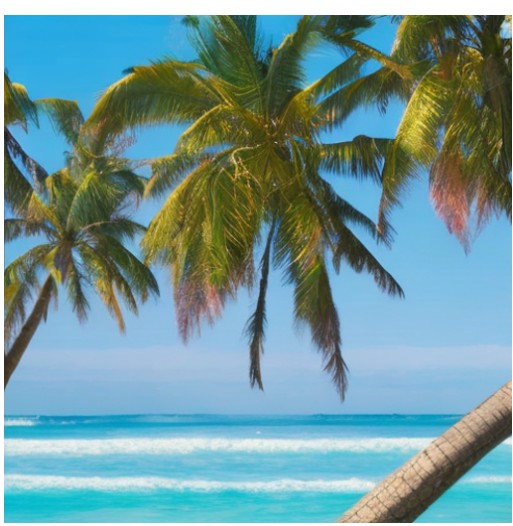

(a) Original Watermarked Image      (b) Watermark Removed Output (Cropping 10px)

Figure 6: Demonstration of watermark removal via cropping attack for prompt generated with border

## A.9 Supporting Proofs

### A.9.1 Proof of Theorem 2

Suppose, for contradiction, that such a code $\Gamma$ exists.

Let $sk \leftarrow_\$ \mathsf{KGen}(1^\lambda)$ and let $\gamma = \mathsf{Enc}(sk)$ be an honestly generated codeword. Consider the tampering function $f$ that acts independently on each coordinate as follows: for each $i \in \{1, \dots, n\}$, set

$$(f(\gamma))_i = \begin{cases} \text{a uniformly random symbol in } \Sigma, & \text{with probability } 1 - 1/q \\ \gamma_i, & \text{with probability } 1/q \end{cases}$$

Let $X$ be the number of positions where $f(\gamma)$ differs from $\gamma$. The random variables $X_i = \mathbf{1}\{(f(\gamma))_i \neq \gamma_i\}$ are independent Bernoulli variables with $\mathbb{E}[X_i] = 1 - 1/q$, so $\mathbb{E}[X] = n(1-1/q)$.

Applying the multiplicative Chernoff bound, for any $\delta \in (0, 1)$,

$$\mathbb{P}[X > (1+\delta)n(1-1/q)] \leq \exp\left(-\frac{\delta^2}{3}n(1-1/q)\right)$$

which is negligible in $n$. Thus, with overwhelming probability, $f$ changes at most $\alpha n = (1 - 1/q)(1+\delta)n$ positions, so $f \in \mathcal{F}_{\alpha n}^{\mathrm{ind}}$ except with negligible probability.

Now, notice that $f(\gamma)$ is distributed exactly as a uniformly random string in $\Sigma^n$. This is because in each position, the symbol is resampled independently and uniformly, so the overall string is independent of $\gamma$.

By the soundness property, for a randomly chosen $\hat\gamma \leftarrow_\$ \Sigma^n$, $\mathbb{P}[\mathsf{Dec}(sk, \hat\gamma) \neq \texttt{invalid}] \leq \mathsf{negl}(\lambda)$.

By tamper detection, for any honestly generated codeword $\gamma$ and any $f \in \mathcal{F}_{\alpha n}^{\mathrm{ind}}$, and for $\tilde\gamma = f(\gamma)$ with $\tilde\gamma \neq \gamma$, $\mathbb{P}[\mathsf{Dec}(sk, \tilde\gamma) \neq \texttt{tampered}] \leq \mathsf{negl}(\lambda)$. But since $f(\gamma)$ is distributed as $\hat\gamma \leftarrow_\$ \Sigma^n$ and is almost always different from $\gamma$, we have for a randomly chosen $\hat\gamma$, $\mathbb{P}[\mathsf{Dec}(sk, \hat\gamma) = \texttt{tampered}] \geq 1 - \mathsf{negl}(\lambda)$.

Therefore, for most random strings $\hat\gamma$, the decoder must output both $\texttt{invalid}$ (by soundness) and $\texttt{tampered}$ (by tamper detection) except with negligible probability. This is impossible, since the decoder can only output one of these values for each input.

We conclude that no messageless secret-key code can be sound and $\mathcal{F}_{\alpha n}^{\mathrm{ind}}$-tamper-detecting at tampering rates $\alpha \geq (1 - 1/q)(1+\delta)$.

### A.9.2 Proof of Theorem 3

We start by describing the messageless secret-key code $\Gamma$. Let $\overline{x} \in \Sigma^*$ be a fixed prompt.

**Key generation $\Gamma.\mathsf{KGen}(1^\lambda)$:** The key-generation algorithm outputs $sk = (\kappa, \mathsf{Model}_\kappa)$ where $(\kappa, \mathsf{Model}_\kappa) \leftarrow_\$ \Pi.\mathsf{Watermark}(\mathsf{Model})$.

**Encoding $\Gamma.\mathsf{Enc}(sk)$:** The encoding algorithm outputs $\gamma = y \in \Sigma^*$ where $y \leftarrow_\$ \mathsf{Model}_\kappa(\overline{x})$.

**Decoding $\Gamma.\mathsf{Dec}(sk, \gamma)$:** The decoding algorithm lets $\gamma = y$ and runs $\Pi.\mathsf{Detect}(\kappa, \overline{x}, y)$. If the result is $\texttt{false}$, it outputs $\texttt{invalid}$. If the result is $\texttt{true}$, it outputs $\texttt{tampered}$.

Note that the decoder never outputs $\texttt{valid}$, and thus $\Gamma$ does not satisfy correctness (which is not required here).

Let us first prove the soundness property. By contradiction, assume $\Gamma$ is not sound. Then, there exists some $\hat\gamma \in \Sigma^*$ such that

$$\mathbb{P}[\Gamma.\mathsf{Dec}(sk, \hat\gamma) \neq \texttt{invalid}] = \mathbb{P}[\Pi.\mathsf{Detect}(\kappa, \overline{x}, \hat\gamma) = \texttt{true}] \geq 1/\mathsf{poly}(\lambda),$$

where the probability is over the choice of the secret key. This violates the soundness of the watermarking scheme.

It remains to prove tamper detection. By contradiction, assume $\Gamma$ does not satisfy $\hat{\mathcal{F}}$-tamper detection. Then, there exists some function $\hat{f} \in \hat{\mathcal{F}}$ such that

$$\mathbb{P}[\Gamma.\mathsf{Dec}(sk, \tilde\gamma) \neq \texttt{tampered} \wedge \tilde\gamma \neq \gamma] = \mathbb{P}[\Pi.\mathsf{Detect}(\kappa, \overline{x}, \tilde\gamma) = \texttt{false} \wedge \tilde\gamma \neq \gamma] \geq 1/\mathsf{poly}(\lambda),$$

where $\tilde{\gamma} \leftarrow \hat{f}(\gamma)$ and where the probability is over the choice of the secret key and the randomness used to generate $\gamma \leftarrow_\$ \mathsf{Model}_\kappa(\overline{x})$. This violates robustness of the watermarking scheme.

