# OpenReview forum: "The Coding Limits of Robust Watermarking for Generative Models"
_ICLR.cc/2026/Conference — ICLR 2026 Conference Withdrawn Submission_

### Official Review · Reviewer_8vog · 2025-10-20

**Soundness:** 2
**Presentation:** 2
**Contribution:** 2
**Rating:** 2
**Confidence:** 5

**Summary:**

The submission provides a study on the limits on the robustness of watermarking for AI generated content.
It models watermarking as the transmission of a secret message over a q-ary alphabet.
The strength of the attack is measured by the number of flipped symbols.

**Strengths:**

I am not sure that the cryptographic formalism is the most suitable for explaining such a simple problem. I would rephrase the study in term of a statistical problem:
- Received sequence in random. What is the probability that the distance wrt the reference sequence is lower than threshold $t$?
- Received sequence is a noisy version of the reference sequence (as modeled Line 1088). What is the probability that the distance gets higher than threshold $t$?
- Can we find a unique threshold satisfying given required upper bounds on these two types of error?

**Weaknesses:**

##W0: Machine Learning
This submission does not deal with AI/ML per se. Right from the beginning, the application is cast as a communication problem where the watermarking primitives are abstracted within a cryptology formalism. Only section 4, the experimental part, considers a practical watermarking technique for GenAI taken on the shelves. I am not sure that this submission falls in the scope of ICLR. I let the decision to the AC.

## W1: Messageless secret-key codes

The introduction of this abstraction is listed as a contribution. I disagree. This is just yet-another wording for naming zero-bit watermarking.
Quoting the authors:
> We introduce and formalize messageless secret-key codes (also referred to as “zero-bit” in prior literature)

So either the difference between messageless and zero-bit is not clearly defined in the submission, either these two are the same, and then , what is the point? The term zero-bit watermarking has been used for more than 20 years.  On the same token, I have difficulty understanding the difference between watermarking and "*cryptographic watermarking*".

## W2: Simple derivations

The proofs rely on a simple derivation based on a Chernoff bound considering two cases:
- $n$ $q$-ary symbols are transmitted and the attack modifies at most $\alpha n$ symbols. The scheme is robust implies that the reception of any sequence closer to the reference codeword than    $\alpha n$ is deemed watermarked. Hence a threshold over this distance.
- If not watermarked, the decoder receives a random sequence. What is the probability that its distance to the referencd codeword is lower than the threshold above mentioned.

Without any surprise, $\alpha < 1 - 1/q$ otherwise attacked sequence and random sequence are statictically indistinguishable.

## W3: Mistake in the proofs

I find the proof not very well written, with some typos or mistakes.
- Theorem 5:
Line 871. It should be $exp(-\frac{\delta^2 n}{3q})$... as the authors did Line 1095.
Line 871. The threshold here equals $t = n - n(1+\delta)/q$, then Line 876, the threshold equals $t=n(1-\frac{1}{q})(1-\delta)$. The footnote (Line 916) mentions a third threshold $t = n(1-\frac{1}{q})(1+\delta)$. Moreover, it states that this third threshold equals the first one. So we end up with three different values for a single threshold:
    - $t_1 = n - n(1+\delta)/q = n(1-1/q-\delta/q)$
    - $t_2 = n(1-\frac{1}{q})(1-\delta) = n(1 -1/q + \delta/q - \delta)$
    - $t_3 = n(1-\frac{1}{q})(1+\delta) = n(1 -1/q - \delta/q + \delta)$
One can tell that: $t_2\leq t_1 < t_3$. Which one is the 'true' threshold?

As for the proof of Th. 2:
The attack replaces a symbol with probability $1-1/q$ by a random symbol. Therefore:
- $E[X_i] = 1/q + (1-1/q)\times 1/q$ (not $1-1/q$ as written Line 1091)
- The attacked sequence is dependent on $\gamma$ (contrary to what Line 1101 says): $P(Y_i=y|\gamma_i)\neq P(Y_i=y)=1/q$.


## W2: Attack on a proposed GenAI watermarking

Section 4 applies the theoretical study to a previous watermarking technique from Gunn et al. which pertains to the family of schemes embedding the watermark in the seed of the diffusion process (like Gaussian Shading, Gauss Shading++). This section shows that this scheme is not robust to a geometric attack (cropping, rescaling, rotation, etc). **This is not a discovery** since the reverse diffusion is not invariant. Indeed, this is already shown in the seminal paper of Gaussian Shading (see Accuracy vs. Random Crop). This is the reason why many schemes (Tree-Rings, Ring-ID, HTSR, Zodiac) seeks robustness against geometric attacks by crafting invariant seeds.

**Questions:**

## Q1
> Line 107: It applies to both secret-key and public-key versions of the primitive

What is public-key watermarking? Can you provide a reference?

## Q2
> Requirements on the Counter.

Moreover, I have serious doubt on the relevance of the proposed scheme in practice: The encoder and decoder must know a secret $\pi$ which is never reused (Line 893). Imagine a watermark detector receiving image I coming from nowhere (ie. a social network or a web page etc). How can it know the associated $\pi$? ... moreover when millions (if not billions) of images are generated each year.

## Q3
> valid,invalid,tampered

I have some doubt about the usefulness of the 'valid' output. At least for watermark technique crafting seeds (as considered in section 4), the reverse diffusion process is noisy, so that even without any attack on the image, the estimated seed is different than the crafted seed.

---

### Official Review · Reviewer_UoE7 · 2025-10-31

**Soundness:** 2
**Presentation:** 2
**Contribution:** 2
**Rating:** 2
**Confidence:** 4

**Summary:**

The paper introduces an abstraction called "messageless secret-key codes" to formalize the requirements for robust cryptographic watermarking in generative models. It proves an information-theoretic threshold for robustness: no scheme can detect tampering if more than half of the bits are modified in binary outputs, or $(1 - 1/q)$ of the symbols for $q$-ary alphabets. The authors provide constructions that approach this bound under cryptographic assumptions (PRFs and a public counter) and demonstrate a practical attack on PRC-watermark using crop-and-resize, which flips about half the latent bits and erases the watermark without visible degradation.

**Strengths:**

1. The abstraction of "messageless secret-key codes" provides a simplified framework for reasoning about some of the core properties of watermarking, such as soundness and tamper detection.

2. The experimental section provides concrete evidence that the theoretical limits may manifest in practice for image watermarking.

3. It connects to prior work on pseudorandom codes (PRCs) and positions the results as establishing optimality for such schemes.

**Weaknesses:**

1. The main theoretical result is a restatement of known information-theoretic limits from classical coding theory. The conclusion that a code cannot be robustly identified if an adversary can modify nearly half of its binary components (or a (1 - 1/q) fraction for q-ary symbols) is a well-understood and intuitive principle from classical coding and information theory. At such an error rate, a tampered codeword becomes statistically indistinguishable from a random, invalid one. While the authors provide a formal proof within their defined abstraction, this feels more like a restatement of a known principle in a new context rather than a surprising or deep new insight for the field of generative model watermarking.

2. The paper's central argument is that the success of the crop-and-resize attack demonstrates a fundamental and universal limit of watermarking itself. This claim seems to be an overstatement resulting from a misattribution of the cause. The attack works because the specific image encoder used (from Stable Diffusion 2.1) produces a radically different latent representation after a seemingly minor crop-and-resize operation. This reveals a weakness in the robustness of the image encoder's representations to geometric transformations, not an inherent flaw in the watermarking method.
The watermark is embedded in the latent space, and its robustness is therefore coupled with the stability of that latent representation. It is entirely plausible that one could train a more robust image encoder, for instance, using data augmentation that includes cropping and resizing, that would produce a much more stable latent vector for the attacked image. In that scenario, the number of flipped latent signs would fall well below the 50% threshold, and the watermark would survive. Therefore, the presented attack is a powerful demonstration of a vulnerability in a specific implementation, but it does not support the broader claim of a universal barrier for all watermarking schemes.

3. The primary cryptographic watermarking schemes (e.g., PRCs) are designed to provide both robustness/error-correction and unforgeability in the multi-message setting. The "messageless" focus unnecessarily reduces the scope of the problem and detracts from analyzing the crucial trade-offs between message length, code rate, and robustness.

**Questions:**

Is the public counter essential in practice? If it is omitted or attacker-controlled, what provable robustness (if any) remains for your constructions?

---

### Official Review · Reviewer_mNLx · 2025-11-03

**Soundness:** 3
**Presentation:** 3
**Contribution:** 2
**Rating:** 6
**Confidence:** 4

**Summary:**

This paper studies the robustness of watermarks embedded in generative models. It introduces a definition of watermarking based on secret-key codes, abstracting the properties of watermarks into soundness, robustness, and pseudorandomness. Based on this definition, it establishes a quantitative threshold for watermark robustness. It provides quantitative values for binary watermarks and q-ary alphabet watermarks, respectively. The paper also conducts robustness experiments on recently proposed watermarking schemes, verifying that operations such as cropping can effectively remove the watermark.

**Strengths:**

-The paper provides a quantitative threshold for the robustness of the watermarking scheme, which facilitates the quantitative analysis of its robustness.

-The paper presents concrete attack experiments on several existing watermarking schemes, providing valuable references for evaluating the security of current schemes.

**Weaknesses:**

Existing research has demonstrated that robustness and soundness of watermarks in generative models cannot be achieved simultaneously. The paper does not cite these findings. Please refer to the specific issues discussed in the next section.

**Questions:**

In the following paper [1], the authors demonstrated that soundness and robustness are conflicting properties, meaning a single watermark scheme cannot simultaneously achieve both. Although the context of [1] focuses on generative text models, which differs from the image scenario addressed in this paper, the underlying principle appears consistent. The authors are requested to clarify whether this alignment holds true. If so, there should be a parameter to adjust the trade-off between robustness and soundness. The conclusion stated in this paper—"For binary outputs, no scheme can survive if more than half of the encoded bits are modified"—does not incorporate such an adjustable parameter. Is this merely a special case?

[1] Two Halves Make a Whole: How to Reconcile Soundness and Robustness in Watermarking for Large Language Models.  https://openreview.net/forum?id=hULJCP47PU

---

### Official Review · Reviewer_s167 · 2025-11-03

**Soundness:** 1
**Presentation:** 2
**Contribution:** 1
**Rating:** 4
**Confidence:** 4

**Summary:**

The paper considers watermarking from an error correction perspective. They show that similar bounds as from error correction apply. They argue that existing schemes already match the limits suggested by these bounds.

**Strengths:**

They argue that existing schemes already reach the theoretical limits in a certain model. It is a nice observation that pseudorandomness can be used in bounding the number of errors that a pseudorandom code can tolerate.

**Weaknesses:**

1. The basic argument about the optimality of the scheme of Gunn, Zhao and Song is that it can tolerate the maximal number of errors. But the number of errors is determined by the randomness recovery algorithm together with the attack! For instance, while cropping may introduce many errors in the particular basis that GZS chose, if one were to e.g. embed the watermark separately in many different sub-images, then it would become robust to cropping. Therefore it is a bit ridiculous to say that it achieves the theoretical limit...

2. The construction does not fit the definition! The construction introduces a counter, which is totally unrealistic. To detect a watermark with a counter, you'd have to iterate over every counter used so far: So if you have generated 1 billion watermarked responses, detection time will be 1 billion times slower than it should be.

3. The claim that this paper gives "A Simplified Abstraction for Watermarking" is not really true at all. What is new about this perspective that wasn't already in all the pseudorandom code papers? For instance, see the introduction of the paper of Christ and Gunn.

**Questions:**

Why do you call it a "messageless secret-key code" instead of a zero-bit pseudorandom code? This is the term already established in the literature, so it is pretty confusing to be re-naming things without explicitly stating why.

---

### Note · Authors · 2025-11-14

I have read and agree with the venue's withdrawal policy on behalf of myself and my co-authors.